# Validation of the Lithuanian Version of the Sociocultural Attitudes towards Appearance Questionnaire-4 (SATAQ-4) in a Student Sample

**DOI:** 10.3390/ijerph17030932

**Published:** 2020-02-03

**Authors:** Migle Baceviciene, Rasa Jankauskiene, Vaiva Balciuniene

**Affiliations:** 1Department of Physical and Social Education, Lithuanian Sports University, Sporto 6, 44221 Kaunas, Lithuania; vaibal@stud.lsu.lt; 2Institute of Sport Science and Innovations, Lithuanian Sports University, Sporto 6, 44221 Kaunas, Lithuania; rasa.jankauskiene@lsu.lt

**Keywords:** SATAQ-4, body image, validity, reliability, factor structure, Lithuanian translation, students

## Abstract

The Sociocultural Attitudes Towards Appearance Questionnaire-4 (SATAQ-4) is one of the most broadly used self-report tools that assess the general role of sociocultural influences on body image and appearance-related internalization. The present study aimed to examine the reliability, validity, and factor structure of the Lithuanian version of the SATAQ-4 (LT-SATAQ-4), as a screening self-report instrument for assessing the role of sociocultural influences on body image. A mixed-gender sample (*N* = 1850) of undergraduate students (88.7%) and graduate students (11.3%) from different state universities and colleges participated in this study (average age 21.6 ±5.0). The students completed a self-report online questionnaire. Intraclass correlation coefficients (ICCs) were calculated for assessing test-retest reliability. The construct validity of the Lithuanian Sociocultural Attitudes Towards Appearance Questionnaire-4 (LT-SATAQ-4) was studied performing exploratory factor analysis (EFA), and then confirmatory factor analysis (CFA). The mean scores for the LT-SATAQ-4 subscales ranged from 1.6 ± 0.9 (Pressure subscale: Peers) to 2.7 ± 1.2 (Internalization subscale: Thin/Low Body Fat). Test-retest reliability was good to excellent for the general and subscale scores (0.85–1.00) except for the Pressure subscale: Peers (0.60). The original 5-factor structure was confirmed by EFA and CFA. Good to excellent internal consistency for each subscale (attempted 0.9 and more) and for the LT-SATAQ-4 global scale (0.91) was obtained. The LT-SATAQ-4 scores had adequate concurrent validity with the measures of the body image, disordered eating, self-esteem, and body mass index. The results support the psychometric properties of the LT-SATAQ-4 and its’ use in Lithuanian student samples. The Lithuanian SATAQ-4 is a useful measure to examine the pressures to internalize appearance ideals in Lithuanian-speaking samples of young individuals.

## 1. Introduction

Eating pathology and body image concerns are major health problems in youth [1]. A plethora of studies were implemented trying to understand the factors involved in the development and maintenance of eating disorders (EDs) and body image concerns [2,3,4,5]. In western cultures, unrealistic beauty ideals are highly valued and are unattainable for the majority of young men and women. Therefore, some of them evaluate themselves as failing to meet social expectations and develop body image concerns related to lower self-esteem, depression, health-compromising eating and exercise behavior [1]. 

Sociocultural factors received major scientific attention, and the tripartite influence model (TIM) was proposed explaining the etiology of body image concerns and disordered eating [2]. This model posits that individuals are pressured by peers, family, and media to adhere to culturally adorable appearance ideals. For girls and young women, the ideal is thin and fit, and for boys and men, it is slightly muscular and low fat [4,6,7]. Research suggests that exposure to idealized media images that emphasize Western values of appearance leads young women and men to body image concerns and eating pathology [3,8,9,10]. Parents, siblings, and family might influence body image of the young women and men through appearance-related comments, appearance-related teasing, appearance-related comments, teasing, or even bullying [11,12]. Taken together, research shows that perceived pressures from media, family, and peers are associated with body dissatisfaction. This relationship works through the internalization of socially adored appearance ideals and social comparisons. Internalization refers to the degree to which an individual “buys into” socially prescribed appearance ideal, and engages in behaviors aimed at meeting these ideals [2]. In the process of social comparison young men and women compare their internalized body image with the actual body and some of them evaluate themselves as failing to meet social expectations developing body dissatisfaction [13]. 

The Sociocultural Attitudes Towards Appearance Questionnaire-4 (SATAQ-4) is one of the most broadly used self-report tools that assess the general role of sociocultural influences on body image and appearance-related internalization [14]. To date, SATAQ-4 has been validated in Italian, Spanish, Portuguese, English, France, Australian, and Japanese samples of young men and women [15,16,17,18] and has demonstrated strong psychometric properties. 

Body dissatisfaction and disordered eating appear significant health problems of young people in Lithuania [19,20,21]. Researches in Lithuania show that body image concerns are prevalent in adolescents and students [19,21,22]. A study in the representative sample demonstrated that every second girl of late adolescent age overestimates her body weight, while more than 60% of boys report body weight underestimation [19]. Unfortunately, there are no data on the prevalence of eating disorders in young women, and men in Lithuania and TIM had never been tested in Lithuanian samples. However, a significant proportion of Lithuanian young women and men report body weight overestimation, body dissatisfaction and health-compromising eating behavior [19,22]. The reduction of body dissatisfaction is one of the essential purposes of prevention programs for disordered eating and body image concerns [23]. Therefore, reliable instruments to measure and evaluate the effects of such interventions in Lithuania are necessary. 

The previous study supported the applicability, factorial validity, and reliability of the SATAQ-3 in the Lithuanian adolescents’ sample [24]. Unfortunately, SATAQ-3 use is limited as it focuses only on media influence without the possibility to test the pressures from peers and family. Further, the SATAQ-3 scale measured a desire to attain a thin physique or physique similar to prominent athletes; however, items do not specifically assess internalization of core physical attributes associated with the thin and/or muscular ideal, i.e., thinness, low body fat, an athletic build or muscularity. These issues were solved in SATAQ-4 [14]. SATAQ-4 is a 22-item questionnaire assessing sociocultural attitudes towards appearance measuring (1) media, family, and peers’ pressure to attain the ideal appearance; (2) internalization of the thin/low body fat; (3) internalization of the muscular/athletic body image. However, SATAQ-4 has not yet been validated using Lithuanian samples. The present study aimed to examine the reliability, validity, and factor structure of the Lithuanian version of the SATAQ-4 (LT-SATAQ-4), as a screening self-report instrument for assessing the general role of sociocultural influences on body image and appearance-related internalization in a Lithuanian student sample. In a current study, we expected that the LT-SATAQ-4 would be deliberated as a stable instrument with adequate concurrent validity, and it would replicate the original 5-factor structure in samples of women and men. 

## 2. Methods

### 2.1. Participants

A mixed-gender sample (*N* = 1850) of 1641 undergraduate (88.7%) and 209 graduate students (11.3%) from different Lithuanian state universities and colleges participated in this study. The sample consisted of 763 male (41.2%) and 1087 female (58.8%) students. The mean age for men was 20.4 ± 3.1 (range 18–46) years; for women, 22.4 ± 5.8 (range 18–42) years. Average body mass index (BMI) for men was 23.4 ± 3.6 (range 15.3–47.3) kg/m^2^; for women 22.1 ± 3.6 (range 14.0–39.8) kg/m^2^.

### 2.2. Procedure

The data were obtained in Lithuanian state universities and colleges during April–October in 2019. The sample of students was from 11 universities and four colleges out of 13 state universities and 12 state colleges in Lithuania. All universities and colleges participating in the present study were located in four major cities of Lithuania. The representativeness of the sample of students was achieved by the compliance of the respondents to the numbers of students in all study areas. Thus, according to the distribution of the general numbers, students in this sample were enrolled in natural and agricultural (6.6%), technology (38.0%), medical and health (27.5%) social and humanities (27.2%) study areas. The researcher V.B. collected the data contacting the administration of the universities and colleges. After gaining oral consent from administrative staff, questionnaires for students were provided.

As part of a more comprehensive study aiming to test associations between students’ body image, health-related lifestyle, and quality of life, the students completed self-report online questionnaires measuring the sociocultural influences, body image, self-esteem, disordered eating, and body mass index (BMI). The students completed questionnaires during scheduled class time, with no time limit. To increase the motivation of the students to complete the survey, an emotional, motivational incentive to enroll in the study was created. The participants received the information that after answering all the questions and answering honestly, they would be authorized to listen remotely to a free four-hour webinar on ‘Healthy Nutrition and Weight Control’. We received 1941 inquiries with 56 refusals to participate in the survey (response rate was 97.1%). The final study sample of 1850 participants with provided complete information was confirmed for statistical analysis.

### 2.3. Ethical Considerations

The researchers received ethical approval to conduct this study by the Committee for Social Sciences Research Ethics of the Lithuanian Sports University (protocol No. SMTEK-7, 13-03-2019). Following the fundamental ethical and legal principles of the research, the students were introduced to the purpose of the study before the questionnaires were presented. The laws of anonymity, goodwill, and volunteering were followed during the survey. To avoid violating national and European Union (EU) legislation, the students were instructed to mark the response “I agree to participate” or “I disagree to participate” to give their consent to participate in the study before beginning the survey.

### 2.4. Measures

Demographic data. Participants in the study were asked to specify their gender, age, type of the higher education institution (university or college), the level of study cycle, study area, study program, and the year of study. 

BMI was based on the self-reported data of the students’ height and weight from which BMI was calculated (kg/m^2^). For sample characteristics, as recommended by the World Health Organization classification, the students’ BMI was classified into four body mass categories: underweight (BMI < 18.5 kg/m^2^), normal weight (BMI = 18.5–24.9 kg/m^2^), overweight (BMI = 25.0–29.9 kg/m^2^) and obese (BMI ≥ 30.0 kg/m^2^) [25]. The majority of the sample (71.7%) was of normal weight. The BMI ranged from 14.0 to 47.3 (M = 22.7, SD = 3.7) kg/m². The results showed that 3.9% of the males and 4.2% of the females were classified as obese (BMI ≥ 30.0 kg/m^2^). A total of 10.3% of the females were underweight (BMI < 18.5 kg/m^2^). Similar weight-related data were observed for the student sample in another representative Lithuanian study [26]. 

The Sociocultural Attitudes Towards Appearance Questionnaire-4 (SATAQ-4) [14] is a 22-item self-report instrument and provides an assessment of the general role of sociocultural influences on body image and appearance-related internalization. The original SATAQ-4 comprise five subscales, each of which is composed of items that are rated on a 5-point Likert scale, where 1 means definite disagreement and 5 means definite agreement. The higher the score, the greater is the acceptance or internalization of the dominant sociocultural standards for appearance. The 5-item (3, 4, 5, 8, and 9) Internalization subscale: Thin/low body fat reveals own’s preference for the appearance of a slender body (e.g., item 3, “I want my body to look very thin.”). The Internalization subscale: Muscular/athletic consists of 5 items (1, 2, 6, 7, and 10) and figures out how much one believes he/she should look muscular and athletic (e.g., item 6, “I spend a lot of time doing things to look more athletic.”). The Pressure subscale: Family indicates family members’ pressure to meet the dominant sociocultural standards for appearance and consists of 4 items (11, 12, 13, and 14; e.g., item 12, “I feel pressure from family members to improve my appearance.”). The 4-item (15, 16, 17, and 18) Pressures subscale: Peers demonstrates the extent to which a person from friends and peers feels pressured to meet social expectations of appearance (e.g., item 16, “I feel pressure from my peers to improve my appearance.”). The Pressures subscale: Media consists of 4-items (19, 20, 21, and 22) and measures the pressures of the media to conform to socially worshiped ideals of body appearance (e.g., item 21, “I feel pressure from the media to improve my appearance.”).

The translation of the SATAQ-4 into Lithuanian was carefully performed by two professional translators and then back-translated to English by two professional translators from a translation agency in Kaunas, Lithuania. The final translation was reviewed by an expert in the field of body image to determine whether the questionnaire covered the concepts it purports to measure. The face validity was rated as good. Finally, a Lithuanian investigator administered the questionnaire to a small group of students and obtained their feedback. This feedback was included in the final Lithuanian translation (Appendix A, Table A1). 

The Lithuanian version of the Multidimensional Body-Self Relations Questionnaire–Appearance Scales (MBSRQ-AS) [27] was employed to assess the appearance-related elements of the body image construct. This instrument of 34 items consists of five subscales, with responses on a 5-point Likert scale ranging from 1 (completely disagree) to 5 (completely agree). The appearance evaluation subscale determines perceptions of physical attractiveness, with a higher score reflecting a higher appearance evaluation, and consists of 7 items (3, 5, 9, 12, 15, 18, and 19; e.g., item 15, “I like the way my clothes fit me.”). The appearance orientation subscale consists of 12 items (1, 2, 6, 7, 10, 11, 13, 14, 16, 17, 20, and21) and reveals the degree of investment in one’s appearance, with a higher score indicating a higher appearance orientation (e.g., item 7, “Before going out, I usually spend a lot of time getting ready.”). The 9-item (from 26 to 34) body area satisfaction subscale assesses satisfaction or dissatisfaction with particular areas of the body (e.g., item 31, “Muscle tone”). A higher score determines greater body area satisfaction. The overweight preoccupation subscale consists of 4 items (4, 8, 22, and 23) and evaluates weight vigilance, dieting, fat anxiety, and eating restraint (e.g., item 8, “I am very conscious of even small changes in my weight.”). A higher score defines a greater preoccupation with being overweight. The 2-item (24 and 25) self-classified weight scale shows how one perceives and identifies one’s weight with a higher score indication of firmer beliefs that bodyweight is too high (e.g., item 25, “From looking at me, most other people would think I am: very underweight/somewhat underweight/normal weight/somewhat overweight /very overweight”. The Lithuanian version of the MBSRQ-AS (LT-MBSRQ-AS) has demonstrated good validity and reliability in a student population sample [22]. The scale was purchased from the author Thomas F. Cash, Ph.D. official site. In the present study, Cronbach’s alpha for the appearance evaluation, appearance orientation, overweight preoccupation, body area satisfaction, and classified weight were 0.83, 0.79, 0.73, 0.88, and 0.85, respectively.

The Lithuanian version of the Eating Disorder Examination Questionnaire 6.0 (EDE-Q 6.0) [28] is a 28-item self-report questionnaire and provides a comprehensive evaluation of the essential behavioral characteristics of EDs and eating disordered behavior. The EDE-Q 6.0 concentrates on the last 28 days and establishes two models of data. First, the six open-ended questions (from 13 to 18) result in frequency data on the essential behavioral characteristics of EDs (number of episodes of the behavior or number of days on which the action has occurred): objective binge eating, self-induced vomiting, laxative use, and excessive exercise (e.g., item 16, “Over the past 28 days, how many times have you made yourself sick (vomit) as a means of controlling your shape or weight?”). Second, 22 attitudinal questions comprise four subscales (the 5-item restraint subscale, the 5-item eating concern subscale, the 8-item shape concern subscale, and the 5-item weight concern subscale) and result in subscale scores that reflect the severity of the ED characteristics. The answer options are arranged on a 7-point Likert scale from 0 (no day) to 6 (every day). A higher score reflects either greater severity or frequency. The Lithuanian version of the EDE-Q 6.0 (LT-EDE-Q 6.0) has demonstrated good validity and reliability in a student population sample [29]. In the present study, Cronbach’s alpha for the LT-EDE-Q 6.0 general was 0.94.

The Lithuanian version of M. Rosenberg’s Self-Esteem Scale (RSES) [30] was used to assess self-esteem and general feelings of self-worth. The scale is composed of 10 items scored on a 4-point Likert scale ranging from 1 (strongly disagree) to 4 (strongly agree). A higher score denotes a greater level of self-esteem. RSES is the most widely used measure of global self-esteem [30]. The tool demonstrated good internal consistency in the present study. Cronbach’s alpha for the RSES was 0.89.

### 2.5. Statistical Analysis

First, descriptive statistics of the sample were performed, and the results are presented as the means ± standard deviations and as percentages according to the type of variable. Normative data for the LT-SATAQ-4 were presented using descriptive statistics. Second, thirty-four volunteer students (all women) were invited to complete the same questionnaire two weeks after they had first completed surveys to investigate the test-retest reliability of the LT-SATAQ-4. Intraclass correlation coefficients (ICCs) were calculated for assessing test-retest reliability. Cronbach’s alpha coefficients were used for the evaluation of internal consistency. A score of ≥0.90 was considered excellent, ≥0.80 good, and ≥0.70 acceptable. Pearson’s correlation coefficients were used for the analyses of construct validity (inter-item correlations and divergent validity). Correlations of the value <0.40 were considered as weak, 0.40–0.59 as moderate, and ≥0.6 as strong. Third, to confirm the concurrent validity, Pearson’s correlation coefficients were used to evaluate the relationships between the subscale scores from the LT-SATAQ-4 and the measures from the LT-MBSRQ-AS, LT-EDEQ-6.0, RSES, and BMI calculations. Next, the construct validity of the LT-SATAQ-4 was studied performing exploratory factor analysis (EFA), and then confirmatory factor analysis (CFA). The sample was randomly divided into two groups. One group was used for EFA (*n* = 950), another one for CFA (*n* = 900). The EFA was performed using the extraction method of principal component analysis with the rotation method of Varimax with Kaiser normalization. Also, using SPSS Syntax File (IBM Corp., Armonk, NY, USA), a parallel analysis was conducted for determining the number of factors in correlation matrices in men and women. Then, using AMOS version 24 (Analysis of Momentary Structure, SPSS; IBM Corp., Armonk, NY, USA), the CFA of the 22-item scale was carried out, and the goodness of fit of the model was assessed using acceptable fit values: the comparative fit index, CFI (0.90 < CFI < 0.95) and the root of the mean square error of approximation, root mean square error of approximation (RMSEA) (0.05 < RMSEA < 0.08). Finally, structural invariance across gender groups of the LT-SATAQ-4 was tested. The statistical analyses were carried out using IBM SPSS Statistics 25 (IBM Corp., Armonk, NY, USA) and AMOS version 24.

## 3. Results

The descriptive statistics for the LT-SATAQ-4 results are presented in Table 1. Items in the scale, median, mean, standard deviation, range, kurtosis, skewness, and percent scoring at the lowest possible value (floor) and the highest possible value (ceiling) were presented to report the statistical characteristics of the instrument. For females, the general scale score was 2.36 ± 0.76; for males 2.07 ± 0.69. For female students Internalization: Thin/low body fat, Internalization: Muscular/athletic, Pressures: Family, Peers and Media means were as follows: 2.97 ± 1.17; 2.63 ± 0.98; 1.79 ± 1.06; 1.54 ± 0.87; 2.70 ± 1.46. For men, accordingly: 2.24 ± 0.96; 2.81 ± 1.04; 1.70 ± 0.94; 1.67 ± 0.95; 1.71 ± 1.09. The skewness and kurtosis coefficients were computed for univariate normality analysis purposes. In men, family, peers and media pressures scores, in women—family and peers pressures scores were positively skewed.

Next, the construct validity of the LT-SATAQ-4 was studied performing EFA and then CFA. The original 5-factor structure was confirmed by EFA in women and men (Table 2 and Table 3). In women, the Kaiser–Meyer–Olkin (KMO) resulted in a measure of sampling adequacy of 0.89, and Bartlett’s test of sphericity (χ^2^ = 11189.2, df = 231, *p* < 0.001) indicated the appropriateness to proceed with exploratory factor analysis. We used the Varimax method to obtain orthogonal factors. Using this method, a 5-factor solution was revealed. The 5-factor model accounted for 78.6% of the total variance. The first factor explained 16.9% of the variance and included items representing Pressures subscale: media. The second factor explained 16.6% of the variance and was defined by the Internalization subscale: Thin/low body fat items while the third by the Internalization subscale: Muscular/athletic statements (15.4%). The fourth factor explained 15.0% of the variance and encompassed items related to Pressures subscale: peers. The fifth factor explained 14.8% of the variance and comprised of statements from Pressures subscale: family. Factors loadings <0.4 in Table 2 and Table 3 are suppressed.

In men, the Kaiser–Meyer–Olkin (KMO) resulted in a measure of sampling adequacy of 0.88, and Bartlett’s test of sphericity (χ^2^ = 8092.4, df = 231, *p* < 0.001) indicated the appropriateness to proceed with exploratory factor analysis (Table 3). We used the Varimax method to obtain orthogonal factors. Using this method, a 5-factor solution was revealed. The 5-factor model accounted for 77.6% of the total variance. For men, variances accounted for each factor were as follows: 18.3%; 17.2%; 14.8%; 13.9% and 13.4%. However, we found that two items from the Internalization subscale: Thin/low body fat exhibited strong loadings onto the expected Internalization: Muscular/athletic subscale (primary loadings 0.42 and 0.66). Item 4 (“I want my body to look like it has little fat”) exhibited low factor loadings in two subscales, and it loaded more strongly (0.58) onto the Internalization: Muscular/athletic subscale, but not onto the expected factor (Internalization: Thin/low body fat. However, item 9 (“I think a lot about having very little body fat”) exhibited a strong loading onto its primary factor (0.66), but cross-loaded (0.45) onto the Internalization: Muscular/athletic subscale. Since the factor loadings of these cross-loaded items were sufficient, we decided to keep them in the final structure of the questionnaire.

The parallel analysis confirmed the five-factor solution in men and women.

The five-factor structure identified via EFA was next evaluated through CFA. The initial CFA indicated poor model fit (Table 4). Invariance analyses across gender groups revealed a statistical difference between unconstrained and a fully constrained model. Increasingly constrained models demonstrated a decrease in model fit. The statistically significant differences were found when testing the assumption about factors loadings, structural covariances, and measurement residuals equalities across genders. 

Next, test-retest reliability, internal consistency, and the level of construct validity (inter-item correlation and divergent validity) of the LT-SATAQ-4 are displayed in Table 5. Test-retest reliability was good to excellent for the general and subscale scores (0.85–1.00) except for the peer pressures subscale (0.60). Cronbach’s alpha for each subscale attempted 0.87 and more in men and women samples, and for the LT-SATAQ-4 global scale was 0.91. The correlations between the items outside the initial subscale were generally weaker than the inter-item correlations.

The concurrent validity of the LT-SATAQ-4 was assessed by testing the associations of the LT-SATAQ-4 scores with the LT-MBSRQ-AS, LT-EDE-Q 6.0, RSES, and BMI measures in men and women (Table 6). The analysis demonstrated these associations in the predicted directions. The LT-SATAQ-4 global score presented the moderate and positive association with the measure of body image for the scores of scales of appearance orientation, overweight preoccupation, and self-classified weight, but for the scores of scales of appearance evaluation and body areas satisfaction obtained negative associations with the LT-SATAQ-4 global scores. LT-SATAQ-4 was related to the LT-EDE-Q 6.0 global score in the expected direction with the Internalization: Thin/low body fat subscale scores demonstrating a high correlation with disordered eating in females and moderate in males. Importantly, Internalization: Muscular/athletic subscale scores were also significantly associated with disordered eating. LT-SATAQ-4 exhibited weak and negative associations with global self-esteem. The correlations between LT-SATAQ-4 subscales and BMI were positive, with the exception of Internalization: Muscular/athletic subscale demonstrating no correlation.

Multiple regression analyses were performed with five LT-SATAQ-4 subscales as the predictive variables and disordered eating as the criterion variable (Table 7). Results revealed that both models were significant (for men F = 61.33; *p* < 0.001; for women F = 208.20; *p* < 0.001), explaining 32.7% of the variance of disordered eating behaviors in men and 53.6% in women. Variance inflation factors (VIFs) ranged from 1.1 to 1.8 in men and from 1.2 to 1.7 in women. Results of the linear regression showed that for women, Internalization: Thin/low body fat and Pressure: family, peers, and media were significantly associated with disordered eating. For men, the findings were the same, except for Pressure: the Family was not associated with disordered eating. Further, Internalization: Muscular/athletic in men was associated with an increased risk of disordered eating behaviors.

## 4. Discussion

The aim of the present study was to examine the reliability, validity, and factor structure of the Lithuanian version of the LT-SATAQ-4, as a screening self-report instrument assessing the general role of sociocultural influences on body image and appearance-related internalization in a nonclinical Lithuanian student sample. We verified the instrument reliability, validity, and factor structure with different psychometric tests. In a current study, we expected that the LT-SATAQ-4 would be deliberated as a stable test with adequate internal consistency, concurrent validity, and it would reflect the original 5-factor structure. 

In general, the SATAQ-4 exhibited good psychometric properties. We found high Cronbach’s alpha coefficients for the scale and five subscales. These results go in line with the originally developed scale [14]. Moreover, the test-retest reliability was good to excellent for the LT-SATAQ-4 global and subscale scores (ICC range was 0.85–1.00) except for the Pressure subscale: peers (ICC was 0.60). This result might be explained by the assumption that the circle of young men’s and women’s peers is changing rapidly in the modern world. 

For women, the factor structure of LT-SATAQ-4 showed a good fit for the original scale structure [14]. The eigenvalues demonstrated that the five-factor model was achieved. However, a slightly different factor structure emerged in men: two items (4 and 9) were cross-loaded between Internalization: Thin/low body weight and Internalization: Muscular/athletic subscales. These findings are in line with other studies demonstrating the problematic acting of some items in men [14,17]. The cross-loadings of the items between the Internalization subscale: Thin/low body fat and the Internalization subscale: Muscular/athletic are not surprising as studies demonstrated that appearance ideals for men emphasize both muscularity and low body fat [4,6]. Since the factor loadings of these cross-loaded items were sufficient, and other studies demonstrate similar results, we decided to keep them in the final structure of the questionnaire.

Analysis of CFA in women demonstrated that according to most fit indices, data fit fairly well. These results are consistent with previous psychometric evaluations of the SATAQ-4 in nationally different female samples [14,15,16,17]. As expected, for men, the model demonstrated unacceptable model fit values. This might be explained by over-loading of items between Internalization: Thin/low body fat and Internalization: Muscular/athletic subscales. According to Schaefer et al., it is possible that independent psychometric testing of all generated items among men and women may produce similar but separate versions of the scale that are responsive to gender-specific appearance concerns [14]. 

LT-SATAQ-4 showed good convergent validity. In line with the findings of the original validation study [14], LT-SATAQ-4 correlated in the expected directions with the measures of body image, disordered eating, and self-esteem. Five subscales showed significant associations with convergent measures. Also, the analyses of linear regressions for women and men demonstrated that Internalization: Thin/low body fat and Pressure: media and peers are associated with disordered eating in women and men. However, in line with other studies, we found that Internalization: Muscular/athletic ideal is significantly associated with disordered eating in men [6]. Next, we observed no direct associations between family pressure and disordered eating in men. However, this finding may indicate that family pressure works through mediators such as internalization of thin/low fat or muscular/athletic ideals, social comparisons, or body image concerns [2]. Future studies testing full TIM models for women and men in Lithuanian samples would expand previously mentioned associations helping to clarify the important mediators between sociocultural pressures and disordered eating. Since this is the instrument validation study, the more profound interpretation of the direct relationships between study variables is limited. 

LT-SATAQ-4 mean scores for thin/Low body fat internalization and pressures scales (family, peers, and media) in women was much lower than in U.S. females, but more similar to non-U.S. females [14]. Mean SATAQ-4 internalization general scale score in female students was also significantly lower than in US women, accordingly: 2.2 ± 0.7 and 3.29 ± 0.92 [5]. However, Internalization: Thin/low body fat and Internalization: Muscular/athletic subscales’ scores were higher than in the Spanish women sample [17]. For men, the mean scores of LT-SATAQ-4 were also lower compared to the U.S. [14]. It might be explained by a lower prevalence of overweight and obesity in Lithuanian students compared to the U.S. [5,31,32]. There is evidence that overweight young people report greater appearance pressures, weight-related teasing, and bullying, and body-image concerns than their counterparts of normal body weight [19,33,34]. Other studies demonstrated lower body image concerns in Lithuanian youth compared to the U.S. [22,35]. 

The previously validated LT-SATAQ-3 has proved a useful instrument for measuring the internalization of media ideals among Lithuanian adolescents of both genders [24]. However, the LT-SATAQ-4 is an updated version of the scale by additionally assessing pressures from families and peers and more detailed internalization of thin/low fat and muscular/athletic ideals. This study adds to the knowledge that educational programs aiming to develop the skills to resist sociocultural pressures for the “ideal” body image and to avoid body image stereotyping are of great importance for young women and men. 

Furthermore, TIM [2] had never been studied in Eastern European youth samples. Therefore, this validated instrument will let us assess significant associations between family, peers, and media pressures, internalization of body ideals, social comparison, body dissatisfaction, and the development of disordered eating. Sociocultural pressures from family, peers, and media for the young people of different sexual identities and BMI might be of very different importance. Moreover, mediators such as internalization of the thin/low body fat or muscular/athletic ideals might play very different roles in the development of body dissatisfaction and disordered eating in young people of different sexual identities and BMI. Future studies testing the TIM models in Lithuanian samples should clarify these issues providing important implications for practice. 

There are several limitations to the present study. First, these data are cross-sectional; therefore, the associations between variables are bidirected and could not allow us to make causal claims. This study examined the sample of student-aged women and men in Lithuania. Therefore, the generalization of the results to other samples of different ages and educational background is limited. Future studies might assess the levels of eating pathology as studies demonstrated different scores of the SATAQ-4 for clinical populations [36]. 

## 5. Conclusions

The results of the current study support the psychometric properties of the LT-SATAQ-4 and its use in Lithuanian student samples. The Lithuanian SATAQ-4 is a useful measure to examine the pressures to internalize appearance ideals in Lithuanian-speaking student-aged samples of men and women.

## Figures and Tables

**Table 1 ijerph-17-00932-t001:** Descriptive statistics of the Lithuanian Sociocultural Attitudes Towards Appearance Questionnaire-4 (LT-SATAQ-4, the Lithuanian student sample *N* = 1850).

Men (*n* = 763)		No. of Items	Median	Mean	SD	Range	Kurtosis	Skewness	Floor (%)	Ceiling (%)
Internalization subscales:	Thin/Low Body Fat	5	2.20	2.24	0.96	1.0–5.0	–0.32	0.52	16.1	1.4
	Muscular/ Athletic	5	3.00	2.81	1.04	1.0–5.0	–0.78	–0.001	6.4	3.0
Pressures subscales:	Family	4	1.25	1.70	0.94	1.0–5.0	1.63	1.45	47.7	1.6
	Peers	4	1.00	1.67	0.95	1.0–5.0	1.29	1.40	54.9	1.0
	Media	4	1.00	1.71	1.09	1.0–5.0	0.93	1.40	61.7	2.6
LT-SATAQ-4 general scale		22	2.00	2.07	0.69	1.0–5.0	0.83	0.73	4.1	0.1
Women (*n* = 1067)										
Internalization subscales:	Thin/Low Body Fat	5	3.00	2.97	1.18	1.0–5.0	–0.92	–0.05	9.1	7.6
	Muscular/ Athletic	5	2.60	2.63	0.98	1.0–5.0	–0.51	0.22	6.7	1.9
Pressures subscales:	Family	4	1.25	1.79	1.06	1.0–5.0	0.83	1.35	46.1	2.0
	Peers	4	1.00	1.54	0.87	1.0–5.0	1.93	1.65	63.0	0.5
	Media	4	3.00	2.70	1.46	1.0–5.0	–1.46	0.11	32.4	11.5
LT-SATAQ-4 general scale		22	2.31	2.36	0.76	1.0–5.0	–0.37	0.30	1.8	0.0

Note: SD = standard deviation; LT-SATAQ-4 global scale = the combined subscales of the Lithuanian version of the Sociocultural Attitudes Towards Appearance Questionnaire – 4.

**Table 2 ijerph-17-00932-t002:** LT-SATAQ-4 factor loadings for the Lithuanian sample of female students (*n* = 561).

LT-SATAQ-4 Items	LT-SATAQ-4 Global Factors and Their Weights				
	Internalization		Pressures		
Item No.	Thin/Low Body Fat	Muscular/Athletic	Family	Peers	Media
Item No. **3**	0.821				
Item No. **4**	0.747				
Item No. **5**	0.799				
Item No. **8**	0.850				
Item No. **9**	0.750				
Item No. **1**		0.721			
Item No. **2**		0.747			
Item No. **6**		0.846			
Item No. **7**		0.756			
Item No. **10**		0.781			
Item No. **11**			0.857		
Item No. **12**			0.871		
Item No. **13**			0.879		
Item No. **14**			0.768		
Item No. **15**				0.757	
Item No. **16**				0.879	
Item No. **17**				0.889	
Item No. **18**				0.873	
Item No. **19**					0.930
Item No. **20**					0.916
Item No. **21**					0.928
Item No. **22**					0.918

Note. Extraction method of principal component analysis. Rotation method of Varimax with Kaiser normalization. Rotation converged in 6 iterations; Kaiser–Meyer–Olkin (KMO) = 0.889.

**Table 3 ijerph-17-00932-t003:** LT-SATAQ-4 factor loadings for the Lithuanian sample of male students (*n* = 389).

LT-SATAQ-4 Items	LT-SATAQ-4 Global Factors and Their Weights				
	Internalization		Pressures		
Item No.	Thin/Low Body Fat	Muscular/Athletic	Family	Peers	Media
Item No. **3**	0.719				
Item No. **4**	0.419	0.581			
Item No. **5**	0.755				
Item No. **8**	0.843				
Item No. **9**	0.662	0.449			
Item No. **1**		0.824			
Item No. **2**		0.785			
Item No. **6**		0.801			
Item No. **7**		0.833			
Item No. **10**		0.796			
Item No. **11**			0.813		
Item No. **12**			0.853		
Item No. **13**			0.789		
Item No. **14**			0.785		
Item No. **15**				0.663	
Item No. **16**				0.858	
Item No. **17**				0.861	
Item No. **18**				0.746	
Item No. **19**					0.918
Item No. **20**					0.881
Item No. **21**					0.917
Item No. **22**					0.888

Note. Extraction method of principal component analysis. Rotation method of Varimax with Kaiser normalization. Rotation converged in 6 iterations; KMO = 0.867.

**Table 4 ijerph-17-00932-t004:** LT-SATAQ-4 confirmatory factor analysis and structural invariance testing across genders.

Models	χ^2^/df	*p*	TLI	CFI	RMSEA
Unconstrained model (general fit across genders)	5.327	<0.001	0.893	0.903	0.068
Constrained models:					
Measurement weights	5.327	<0.001	0.893	0.903	0.067
Structural covariances	5.318	<0.001	0.893	0.900	0.068
Measurement residuals	5.519	<0.001	0.888	0.890	0.070
Men (*n* = 374)	4.976	<0.001	0.879	0.896	0.103
Women (*n* = 526)	5.346	<0.001	0.909	0.922	0.092

TLI = Tucker-Lewis Index; CFI = comparative fit index; RMSEA = root mean square error of approximation.

**Table 5 ijerph-17-00932-t005:** Reliability and validity of the LT-SATAQ-4 (the Lithuanian student sample I = 1850).

LT-SATAQ-4 Subscales		Test-Retest Reliability (ICC)	Cronbach’s α		Inter-Item Correlation *		Divergent Validity **	
			Men	Women	Men	Women	Men	Women
**Internalization subscales:**	Thin/Low Body Fat	0.89	0.870	0.914	0.577	0.681	0.288	0.296
	Muscular/Athletic	0.85	0.898	0.867	0.636	0.565	0.187	0.197
Pressures subscales:	Family	1.00	0.886	0.911	0.682	0.729	0.274	0.225
	Peers	0.60	0.923	0.933	0.750	0.778	0.314	0.275
	Media	0.89	0.970	0.974	0.891	0.905	0.277	0.255
LT-SATAQ-4 general scale		0.88	0.915	0.914	0.335	0.330	-	-

Note: ICC = intraclass correlation coefficient; * Mean value of Pearson correlations coefficients between items within the assigned subscale; ** Mean value of Pearson correlations coefficients between items in subscales other than their own; LT-SATAQ-4 general scale = the combined subscales of the Lithuanian version of the Sociocultural Attitudes Towards Appearance Questionnaire − 4.

**Table 6 ijerph-17-00932-t006:** Correlations between the LT-SATAQ-4, LT-MBSRQ-AS, LT-EDE-Q 6.0, RSES scores, and BMI measures.

Subscales	LT-SATAQ-4					
	Internalization		Pressures			
Men (*n* = 763)	Thin/Low Body Fat	Muscular /Athletic	Family	Peers	Media	LT-SATAQ-4 global
LT-MBSRQ-AS						
Appearance evaluation	−0.191 **	0.118 *	−0.214 **	−0.209 **	−0.162 **	−0.172 **
Appearance orientation	0.262 **	0.446 **	0.071	0.131 **	0.176 **	0.338 **
Overweight preoccupation	0.516 **	0.349 **	0.323 **	0.382 **	0.348 **	0.560 **
Body areas satisfaction	−0.195 **	−0.005	−0.192 **	−0.214 **	−0.205 **	−0.224 **
Self-classified weight	0.314 **	−0.015	0.228 **	0.214 **	0.148 **	0.247 **
LT-EDE-Q 6.0	0.424 **	0.282 **	0.251 **	0.368 **	0.351 **	0.487 **
RSES	−0.112 **	0.012	−0.106 **	−0.143 **	−0.175 **	−0.144 **
BMI	0.213 **	0.021	0.177 **	0.169 **	0.153 **	0.205 **
Women (*n* = 1087)						
LT-MBSRQ-AS						
Appearance evaluation	−0.429 **	−0.078*	−0.384 **	−0.339 **	−0.270 **	−0.438 **
Appearance orientation	0.374 **	0.311 **	0.091 **	0.070 *	0.192 **	0.329 **
Overweight preoccupation	0.633 **	0.383 **	0.365 **	0.346 **	0.329 **	0.618 **
Body areas satisfaction	−0.402 **	−0.099 **	−0.345 **	−0.305 **	−0.307 **	−0.431 **
Self-classified weight	0.400 **	0.093 **	0.430 **	0.322 **	0.272 **	0.441 **
LT-EDE-Q 6.0	0.622 **	0.278 **	0.460 **	0.438 **	0.416 **	0.657 **
RSES	−0.256 **	−0.079 **	−0.195 **	−0.235 **	−0.188 **	−0.279 **
BMI	0.223 **	0.016	0.389 **	0.276 **	0.200 **	0.311 **

Note: * *p* < 0.05; ** *p* < 0.01; LT-SATAQ-4 = Lithuanian version of the Sociocultural Attitudes Towards Appearance Questionnaire–4; LT-MBSRQ-AS = Lithuanian version of the Multidimensional Body-Self Relations Questionnaire–Appearance Scales; LT-EDE-Q 6.0 = Lithuanian version of the Eating Disorders Examination Questionnaire–6.0; RSES = M. Rosenberg Self-Esteem Scale; BMI = body mass index.

**Table 7 ijerph-17-00932-t007:** Predicting LT-EDE-Q 6.0 scores from LT-SATAQ-4 subscales in a students’ samples of men and women (*n* = 1850).

Predictors	Men (*n* = 763)			Women (*n* = 1087)		
	B	β	*p*	B	β	*p*
Body mass index	0.063	0.271	<0.001	0.072	0.210	<0.001
LT-SATAQ-4: Internalization Thin/Low Body Fat	0.183	0.210	<0.001	0.501	0.474	<0.001
LT-SATAQ-4: Internalization Athletic/Muscular	0.086	0.107	0.004	−0.055	−0.044	0.08
LT-SATAQ-4: Pressures, Family	−0.049	−0.055	0.153	0.141	0.121	<0.001
LT-SATAQ-4: Pressures, Peers	0.156	0.177	<0.001	0.196	0.137	<0.001
LT-SATAQ-4: Pressures, Media	0.128	0.166	<0.001	0.100	0.117	<0.001
Model summary	R = 0.572; R^2^ = 0.327			R = 0.732; R^2^ = 0.536		

B—regression coefficient, β—standardized regression coefficient; LT-SATAQ-4—Lithuanian Social Attitudes Towards Appearance Questionnaire – 4.

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
