# Peer review of "Validation of the Lithuanian Version of the Sociocultural Attitudes towards Appearance Questionnaire-4 (SATAQ-4) in a Student Sample"

_ijerph, 2020, doi:10.3390/ijerph17030932_

Round 1

Reviewer 1 Report

The manuscript aims to create and validate a Lithuanian version of SATAQ-4. The objective is very meaningful because the scale has been translated and examined in various cultures. Having different versions of the scale will make it possible to compare sociocultural influences across various cultures. However, there are several issues that I need to address. First of all, there are some grammatical errors throughout the manuscript. Not many, but the authors may want a native English speaker to proofread the manuscript. Secondly, there are some punctuation and format errors as well (spaces are missing, commas and periods are mixed up, etc.). Again, the authors need to pay close attention to such details. Thirdly, SATAQ-4 subscale names are not consistent throughout the manuscript, and sometimes subscale names are different from the original ones. The authors may stick to the original subscale names. Other issues are described below for each section. “L. #” indicates a line number.

[Abstract]

Abbreviations are used for two scales (EDE-Q and RSES). As they appear for the first time in the manuscript, it may be reader-friendly if their full-names are stated. The last sentence is slightly difficult to understand.

[Introduction]

L. 28-30: The authors state that the beauty ideals of Western women and men are unrealistically thin, but the ideals of Western men are highly muscular, which is actually stated by the authors in the following paragraph. If the authors mean that thin women are glorified or well-liked by both women and men, then the authors may rephrase the sentence to clarify it. Sometimes the authors write “throw” instead of “through” (e.g., L. 41, 43). L. 39-42: How peers and family members influence one’s body image seem almost the same, so it can probably be described in one sentence. L. 39, 41: Moreover, the authors write that peers and family members pressure “students” to adhere to appearance ideals whereas the media pressure “young women and men.” I suppose the authors can use a general term for pressure from peers and family members as well rather than using a specific term (“students”). L. 47-50: SATAQ-4 was also validated in Japanese samples (both women and men). L. 50: When body-image researchers talk about how body dissatisfaction influences pathological eating, it is usually discussed in relation to eating disorders (i.e., anorexia nervosa and bulimia nervosa). However, the authors mention obesity only. Is obesity the major eating problem related to body dissatisfaction in Lithuania? L. 57: The authors state that SATAQ-3 did not assess athletic ideals, but it actually did. It is just that “athletic ideals” in SATAQ-3 were ambiguous because some people might imagine gymnasts whereas others might imagine weight-lifters. The authors may clarify this point.

[Methods]

L. 70: The authors can delete “students” after parentheses. L. 71: Currently, the proportions of male and female participants are presented in parentheses before the words “male” and “female.” However, it appears rather unsmooth. I recommend that the authors present numbers outside parentheses and percentages in parentheses (e.g., “763 male students (41.2%) and 1087 female students (58.8%) …” or something like that). L. 76-77: The authors state that 15 schools were “selected” out of 25 schools, but how and why were these schools “selected?” L. 77: The authors write “As part of a more extensive study,” but more extensive than what? L. 81-82: The authors state that “…an emotional, motivational incentive to enroll in the study was created,” but how was it created? Did the authors offer something? L. 97-104: The authors categorize participants’ weight status based on WHO classification, which is nice, but if the authors can also include weight-related data of Lithuania such as average BMI (in comparison to the participants’ weight-related data), that may be nice, too. Readers may want to know if the current sample is representative of “typical” young Lithuanians. L. 112-116: The authors use the word “pressure” to describe what pressure subscales assess, but they may select a different word. If the authors include an example of each scale/subscale items, readers can better understand what each scale/subscale assesses. L. 165-166: The authors classify correlation-coefficient values, but from where the classification schema comes is not clearly stated. L. 174-176: The authors use certain indices to test the goodness of fit of a model, but it is unclear how the certain indices and values are determined. It seems that the authors conducted EFA and CFA on the same sample, but it is not an optimal method. I suggest that the authors randomly split the sample into two groups and run EFA on one group, then CFA on another group. Recently, it is suggested to run a parallel test and/or revised Valicer’s MAP test to determine the number of factors.

[Results]

In Table 1, the range of each SATAQ-4 subscale score is reported as “4.0,” but a range is usually reported as “X to Y.” What does “4.0” mean? Also, if the authors can separately report descriptive stats of male and female participants in a table, that will be nice as readers can compare the data with the data of other samples. The authors do not specify the rules of cross-loadings or low loadings before running factor analyses. That is, how cross-loadings and low loadings will be determined and how they will be treated if they are detected are not specified. In relation to the abovementioned issue, the authors say that two items showed cross-loadings (only for male participants) but they do not clearly say anything about how they were treated as a result (i.e., if and why they were kept or deleted are not clearly reported). The authors find that Pressures: Peers subscale have relatively low test-retest reliability compared to other subscales, but do not speculate why. They may be able to state the possible reason(s) for this finding (in the Discussion section, after L. 289).

[Discussion]

L. 302: Schaefer is misspelled as Shaefer. L. 319: The citation is given in parentheses, not in brackets. I suppose it is [35]. L. 322: The authors report that there are some differences in average scores of SATAQ-4 subscales compared to samples of other nations and say that reasons are probably related to sociocultural differences. However, the authors do not describe specific sociocultural differences in details. L. 330-332: Social comparison and body dissatisfaction are important components of Tripartite Influence Model but they are omitted here. I wonder if the schools from which participants were derived are located in an urban or rural area. Those in urban and rural areas tend to experience different degrees and types of sociocultural influences.

Author Response

Dear Reviewer,

Thank you very much for the precise review which helped us to improve our manuscript. Please find the comments and answers table attached.

Reviewer 2 Report

This paper examined the factor structure, reliability and concurrent validity of a Lithuanian version of the SATAQ 4 scale in a student sample (men and women). The original five factor structure of the scale has been replicated. Overall this is an interesting study. The findings will be helpful to those wishing to use a Lithuanian version of the scale and potentially useful in furthering the understanding of sociocultural attitudes towards appearance across cultures. Overall the study described was well conducted; however, there are some issues that should be addressed throughout the manuscript.

Introduction

The introduction would be improved with the inclusion of more cultural rational for the importance of validating the LT-SATAQ 4. Specifically, the authors should provide more information about the associations between sociocultural influences (e.g. family pressure, peer influences, and muscular and thin internalization) and body dissatisfaction among non-Western/Lithuanian sample of male and female young people. In addition, the authors could provide a brief definition of “internalization” and “social comparison” constructs.

Method

Participants. It could be helpful to provide more information about demographic characteristics of the sample (e.g. mean age, mean BMI, age range and BMI range separately for men and women).

Measures. Please, provide more information about each measurement instrument. Specifically, for each scale you used add a sample item, and for MBSRQ-AS add information about the number of items for each subscale.

Procedure. It could be useful to add some information about the way in which the sample for investigating the test-retest reliability has been selected.

Results

The authors do not mention any exploration of the missing values, or type of missingness, some information about this are needed.

A more articulated description of the findings (i.e. concurrent validity, test-retest reliability, Cronbach’s alpha, construct validity) for men and women separately would be helpful here.

Discussion

The discussion would be improved with a deeper reflection about the association between sociocultural influences and body dissatisfaction/eating disorders among Lithuanian young men and women.

On line 311-313, I would like to see a more forthcoming discussion of the negative correlations between the Internalization: Athletic/Muscular subscales and disordered eating among women.

On line 322, the authors say that their findings could “be explained by sociocultural differences” between Lithuanian and US students: which type of sociocultural differences did they refer to?

I suggest authors to better discuss the implications of their findings to practice; for example, it could be useful to better explain how the findings could be used by practitioners for “body image concern and eating disorders prevention programs in Lithuania” (lines 332-333)

Author Response

Dear Reviewer,

Thank you very much for your comments and remarks. All the corrections made in the paper are highlighted. Please find the comments and answers table attached.
